# Artificial gauge fields and dimensions in a polariton hofstadter ladder

Simon Widmann [1,2] ✉, Jonas Bellmann[1,2], Johannes Düreth [1,2], Siddhartha Dam [1,2], Christian G. Mayer [1,2], Philipp Gagel[1,2], Simon Betzold [1,2], Monika Emmerling[1,2], Subhaskar Mandal [3,4], Rimi Banerjee[3], Timothy C. H. Liew [3], Ronny Thomale [2,5], Sven Höfling[1,2] & Sebastian Klembt [1,2,6] ✉

Artificial gauge fields allow uncharged particles such as photons to mimic the behaviour of charged particles subjected to magnetic fields, providing a powerful platform to alter the effective dynamics and exploring topological physics. Topological exciton-polariton lasers have attracted considerable interest, but often require strong magnetic fields to realise propagating topological edge states. Here we experimentally realise the topological Hall effect in a micron-scale micropillar chain by using an artificial gauge field, exploiting the circular polarisation of polaritons as an artificial dimension. Careful rotational alignment of elliptical micropillars induces strictly polarisation-dependent edge-state propagation, demonstrating non-reciprocal transport of the polariton pseudospins. Our results demonstrate that the dimensionality limitation of topological interface states as well as requirements for strong external magnetic fields in coupled topological laser arrays can be overcome. Our results open new ways towards the implementation of topological polariton lattices and related optically active devices with additional artificial dimension.

Topology has emerged as a complex yet powerful new tool to control and manipulate the flow of light[1-4] as well as tailor the interaction of laser arrays[5-9]. Time-reversal symmetry can be broken by means of strong magnetic fields in electronic[10] or coupled light-matter systems[11]. The electric charge governs the strength and the direction of a particle's response to external electromagnetic fields. However, particles without a charge, such as photons, are largely unaffected by these fields. Artificial gauge fields address this limitation by harnessing geometric effects or modifying the properties of the surrounding medium, causing uncharged particles to mimic the behaviour of charged ones under an external field. Synthetic gauge fields can bestow systems with a variety of intriguing properties normally not found in nature. This concept is particularly valuable in photonics, where it enables synthetic magnetic effects in a domain typically devoid of intrinsic magnetic responses[3,4,6,12]. Such fields form the basis of novel wave-guiding mechanisms and topological phenomena, offering new opportunities for exploring physics and developing innovative devices. In photonics, synthetic gauge fields are pivotal for realising topological phenomena[13,14]. Notable examples include the first photonic topological insulator that utilised helical Floquet waveguides[2,12], photonic TIs and lasers using tailored coupler delay lines between coupled ring resonator structures[3,4,6] as well as the first topological insulator in synthetic dimensions[15]. The incorporation of artificial dimensions has garnered substantial interest as it allows for

[1]Technische Physik, Universität Würzburg, Am Hubland, Würzburg, Germany. [2]Würzburg-Dresden Cluster of Excellence ct.qmat, Universität Würzburg, Am Hubland, Würzburg, Germany. [3]Division of Physics, School of Physical and Mathematical Sciences, Nanyang Technological University, Singapore, Singapore. [4]Department of Physics, Indian Institute of Technology Bombay, Mumbai, India. [5]Lehrstuhl für Theoretische Physik 1, Universität Würzburg, Am Hubland, Würzburg, Germany. [6]Present address: Lehrstuhl für Experimentelle Physik 1, Universität Würzburg, Am Hubland, Würzburg, Germany. ✉e-mail: simon.widmann@uni-wuerzburg.de; sebastian.klembt@uni-wuerzburg.de

increasing the dimensionality of photonic lattices (even beyond 3D[16,17]) and facilitates tailored long-range couplings.

Modern fabrication methods have allowed for the realisation of semiconductor microcavities with very high quality factors. When quantum wells are embedded within these cavities, the likelihood of a photon re-exciting a quantum-well exciton can surpass its probability of leaving the cavity. In this strong coupling regime, exciton-polaritons (or simply polaritons) emerge as hybrid light-matter quasiparticles, exhibiting properties of quantum fluids of light[18]. Innovative cavity designs and geometries have further allowed the confinement of polaritons within one- and two-dimensional potential landscapes[19], leading to realisations of topological polariton insulators[11,20,21] and Hamiltonian simulators[22]. By imposing periodic in-plane potentials, exciton-polariton lattices can be created, mirroring many properties of microscopic crystalline solids. Here, micropillars act as analogs to atoms: small resonators patterned within planar cavities, typically circular in shape, exhibit discrete energy levels with eigenvectors that resemble hydrogen-like ($s, p, d,...$) orbitals. When these micropillars are closely and periodically spaced, their previously localised individual pillar eigenstates overlap. In this manner, band structure formation can be observed and controlled with a large range of tunable parameters. At sufficiently high densities, these Bosonic quasiparticles can undergo a transition into a driven-dissipative Bose–Einstein condensate-like state with macroscopic phase coherence[23]. If condensation occurs in a state with finite group velocity, exciton-polariton condensates can propagate, e.g. in a waveguide or in the edge mode of a polariton lattice[11].

In this work, the microcavity is designed such that polaritons are confined within a linear chain of coupled elliptical micropillars. The elliptical shape of the micropillars introduces anisotropy, lifting the degeneracy of the fundamental mode and resulting in two linearly polarised modes aligned with the major and minor axes of the ellipse. By systematically rotating each elliptical pillar relative to its neighbours, we tailor the coupling between these modes, thereby implementing an artificial gauge field in a compact micron-scale optically active device. This rotation imposes a phase accumulated by polaritons as they traverse a plaquette within the effectively two-dimensional polariton lattice, emulating the dynamics of charged particles in an external magnetic field. The synthetic dimension is created by this tailored potential landscape, facilitating the investigation of topological phenomena in two-dimensional space in a linear micropillar chain.

## Results

### Hofstadter Hamiltonian in an elliptical micropillar chain

To capture the dynamics of our system, we employ a tight-binding model that describes polaritons in a lattice potential. This model provides a foundation for understanding the behaviour of our polaritonic lattice, which shares key properties. The rotation of neighbouring elliptical micropillars introduces complex hopping amplitudes in the direction of the artificial dimension, analogous to the phase accumulation of charged particles in a real magnetic field. We begin by considering a simpler, illustrative system: a charged particle on a two-dimensional square lattice subjected to a uniform external perpendicular magnetic field. Such a lattice was initially theorised by Harper[24], but owes its name to the subsequent discovery of the butterfly-like eigenvalue structure by Hofstadter[25]. It was later discovered that the system plays an important role in understanding the quantised Hall conductance in the quantum Hall effect and that its topology can be classified by a Chern number[10,26,27]. In this Hofstadter Hamiltonian, we specifically analyse a lattice that is periodic along the $x$-direction and comprises $N_y$ sites along the $y$-direction (Fig. 1(a)). The hopping along the $x$-direction is characterised by a real amplitude $J$, while the hopping along the $y$-direction is described by a complex amplitude $\Delta e^{in_x\phi}$ in the Landau gauge[25]. The phase $\phi$ depends on the lattice site index $n_x$ in the

$x$-direction. This phase factor encodes the effect of the magnetic flux through a plaquette. When a particle encircles a plaquette clockwise, it accumulates a phase $\phi$. By setting $\phi = 2\pi/3$, the magnetic field introduces a periodicity of three lattice sites in the $x$-direction, such that the unit cell repeats after $n'_x = n_x + 3$. The tight-binding Hamiltonian describing this system is given by[28]

$$\hat{H} = \sum_{n_x, n_y} -[J\hat{a}^\dagger_{n_x, n_y}\hat{a}_{n_x+1, n_y} + \Delta e^{in_x\phi}\hat{a}^\dagger_{n_x, n_y}\hat{a}_{n_x, n_y+1}] + \text{h.c.} . \quad (1)$$

The resulting band structure reveals three bulk bands connected by topological edge states (Fig. 1b).

Reducing the lattice size along the $y$-direction to just $N_y = 2$ sites removes these bulk bands while preserving the edge states in this Hofstadter ladder. The edge localisation is further illustrated by the expectation value of the edge localisation operator $\hat{L}$.

In the reduced lattice with $N_y = 2$, the lower and upper edges can be associated with the left ($\psi_{\sigma-}$) and right ($\psi_{\sigma+}$) circularly polarised polariton spin states of an elliptical micropillar chain. Elliptical micropillars inherently exhibit polarisation splitting of their fundamental mode, resulting in two linearly polarised modes aligned along the longitudinal (semi-major) and transverse (minor) axes of the ellipse, denoted as $\psi_L$ and $\psi_T$[29]. These linearly polarised modes can be re-expressed in the circular polarisation basis as $\psi_{\sigma\pm} = (\psi_L \pm i\psi_T)/\sqrt{2}$. In this system, the coupling $J$ corresponds to the hopping between neighbouring pillars, while $\Delta$ represents the intrinsic polarisation splitting of the fundamental modes within the individual micropillars[30]. The phase factor $\Delta e^{in_x\phi}$, responsible for the $2\pi/3$ plaquette phase, is implemented through a deliberate rotation of the micropillars. By rotating each micropillar by an angle $\theta = \phi/2$, a net phase of $e^{in_x\phi}$ is induced in the coupling between the circularly polarised eigenstates within a micropillar. This design realises the synthetic magnetic flux through the plaquettes. A detailed explanation of this complex hopping mechanism is provided in the methods section[31].

A schematic of the elliptical micropillar chain, including different refractive index layers and the optically active region, is displayed in Fig. 1c (not to scale). The microcavity structure comprises quantum wells embedded within a cavity, sandwiched between highly reflective ($R > 99.9\%$) top and bottom distributed Bragg reflectors (DBRs) to achieve strong optical confinement. The crystalline structure is grown using molecular beam epitaxy on the gallium arsenide III-V semiconductor platform. A more precise overview of the planar structure is given in the methods section. Patterning of the planar microcavity into elliptical micropillars is achieved using an electron-sensitive photoresist, electron beam lithography, and subsequent inductively coupled plasma etching. This process allows for a high etch-depth aspect ratio and low sidewall roughness, both of which are essential for maintaining optical performance. The etching extends through the cavity layer and partially into the bottom DBR to create well-defined micropillars with sufficient confinement to achieve the desired linear polarisation-induced splitting of the fundamental mode of the elliptical micropillars. To enhance both mechanical and chemical stability, the etched structures are spin coated with benzocyclobutene (BCB), a transparent and low-refractive-index polymer (Fig. 1d). This coating improves durability while maintaining optical integrity. Figure 1e shows the three-ellipse unit cell with $\theta = \pi/3 = 60°$ rotations that enforce the three-site periodicity.

The strong coupling regime in the microcavity is verified through the observed anti-crossing of the eigenstates in Fig. 2a, measured via white-light reflection on a planar part of the microcavity. The wavevector component perpendicular to the cavity, $k_z$, is inversely related to the cavity length, $k_z = \pi \cdot L_z^{-1}$. The spectra of the anticrossing in Fig. 2a have a linewidth that is significantly narrower than the energy separation of the modes. To further investigate the input-output characteristics of the system, non-resonant continuous-wave laser

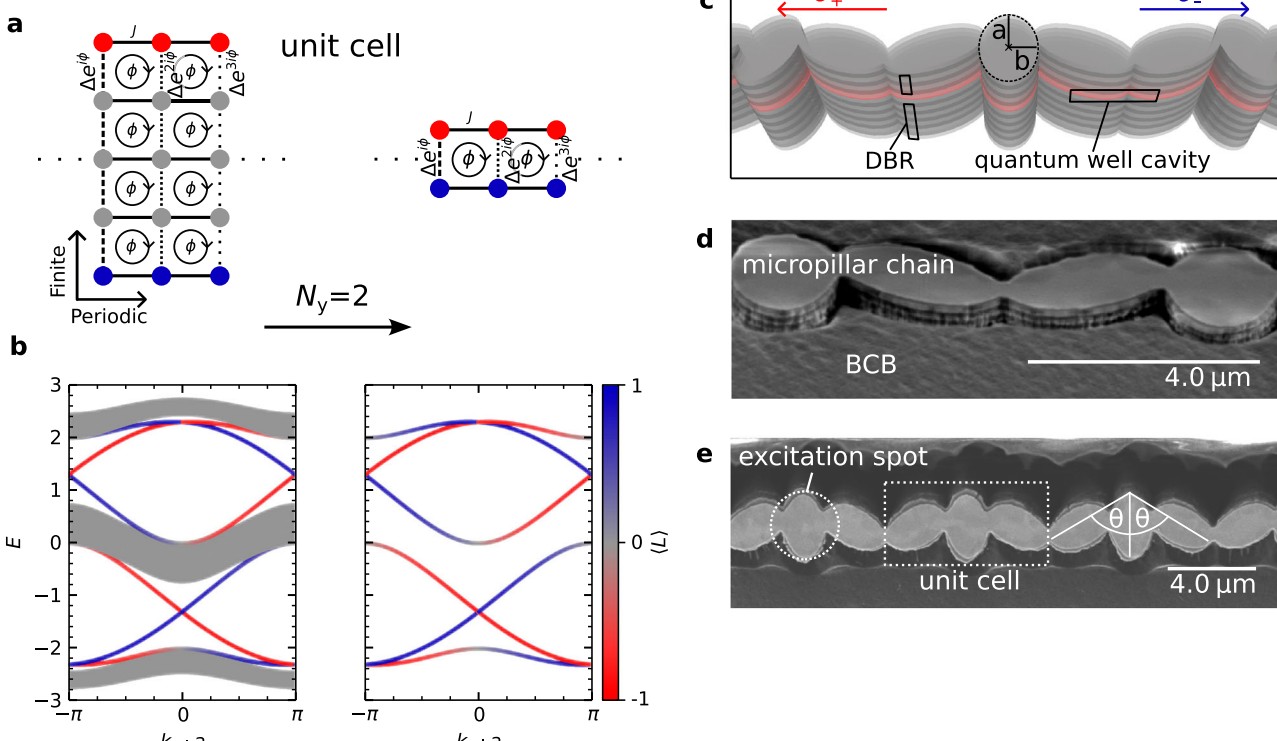

**Fig. 1 | Hamiltonian and system geometry. a** Unit cell of the two-dimensional square lattice described by the Hofstadter Hamiltonian[25]. The lattice exhibits real hopping $J$ in the $x$-direction and complex hopping $\Delta e^{in_x\phi}$ in the $y$-direction, where $n_x$ is the lattice position along $x$. With $\phi = 2\pi/3$, the system has a three-lattice-site periodicity in the $x$-direction, where periodic boundary conditions are applied. Edge sites along the finite $y$-direction are colour-coded red and blue. **b** Band structure of the system, overlaid with the edge localisation operator $\langle \hat{L} \rangle$, which takes the values of 1 (−1) for complete localisation at the lower (upper) edge. For $N_y = 2$, the bulk bands vanish, leaving only the topological edge states. **c** Schematic of the structured microcavity, showing the Bragg mirrors and the optically active cavity layer. Circularly polarised polaritons propagate in opposite directions within the cavity. **d** Scanning electron microscope (SEM) close-up of the fabricated structure surrounded by the protective polymer benzocyclobutene (BCB). The micropillar chain consists of overlapping ellipses rotated by 120° with respect to each other. **e** SEM image showing the unit cell that contains three ellipses. The three-pillar periodicity results from the $\theta = \pi/3 = 60°$ rotation angle.

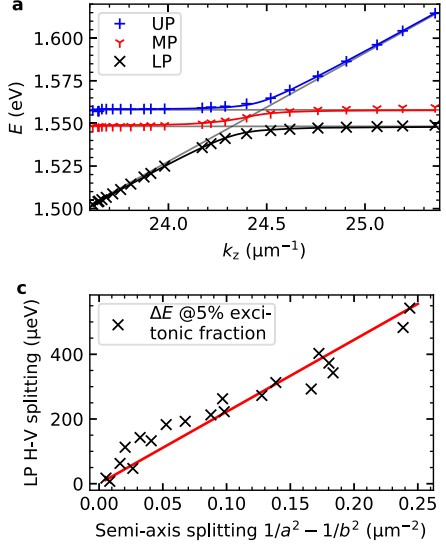

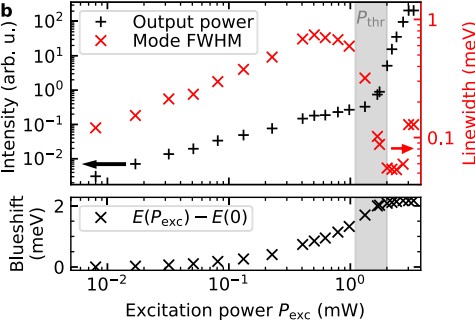

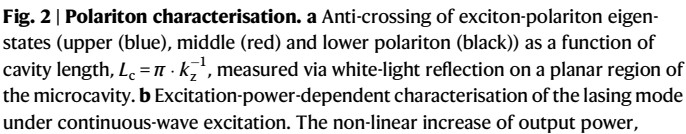

**Fig. 2 | Polariton characterisation. a** Anti-crossing of exciton-polariton eigenstates (upper (blue), middle (red) and lower polariton (black)) as a function of cavity length, $L_c = \pi \cdot k_z^{-1}$, measured via white-light reflection on a planar region of the microcavity. **b** Excitation-power-dependent characterisation of the lasing mode under continuous-wave excitation. The non-linear increase of output power, linewidth narrowing, and blueshift exhibit typical polariton lasing behaviour. **c** Energy splitting of the fundamental mode in elliptical micropillars for varying ellipticities and sizes. The fundamental mode energy is inversely proportional to the square of the ellipse semi-axes.

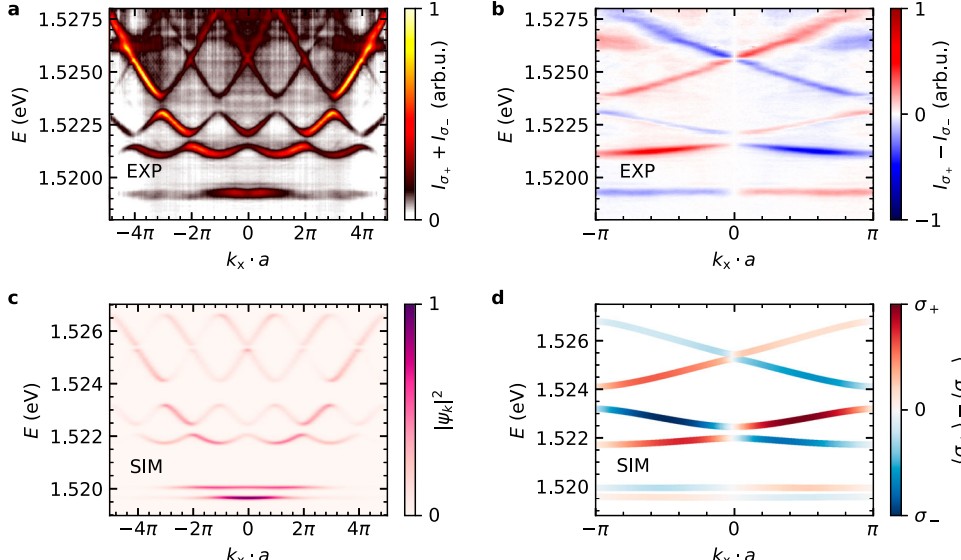

**Fig. 3 | Experimental and theoretical band structures of the elliptical micropillar chain. a** Angle-resolved emission spectrum of the elliptical micropillar chain with a lattice constant $a$. **b** Difference between $\sigma_\pm$ polarisation-resolved band structures. Due to experimental limitations, optimal circular polarisation detection is only achievable for $k_x = 0$, with reduced detection efficiency at larger $k_x$. Therefore, only the first Brillouin zone is depicted. To further address mirror

symmetry breaking $k_x \rightarrow -k_x$ introduced by the detection setup, the spectrum is symmetrized around $k_x = 0$. The unsymmetrized spectrum is displayed in Fig. 6 in the methods section. **c** Simulated band structure of a finite micropillar chain, demonstrating agreement with experimental data. **d** Polarisation-resolved band structure of an infinite micropillar chain.

excitation is performed on a single micropillar of the chain (Fig. 1e) at the first higher energy Bragg minimum of the microcavity. The input-output characteristics under these conditions reveal hallmark features of exciton-polaritons: a nonlinear increase in output power, linewidth narrowing, and a blueshift at a threshold power of approximately $P_{thr} = 1.3$ mW. These dependencies are summarised in Fig. 2b. These results demonstrate the preservation of strong coupling in the patterned elliptical micropillars. By employing a fully etched patterning technique, the elliptical geometry induces a linear polarisation splitting of the fundamental mode of up to 0.5 meV, as shown in Fig. 2c. This splitting arises from the different linear polarisation dependent continuity conditions of the electric field and the electric displacement field at the edges of the elliptical micropillars, where the refractive index changes (semiconductor to BCB). Mathematically, the splitting in individual ellipses is closely related to a particle in a two-dimensional box potential. There, the energy difference, $\Delta E$, follows the relationship $\Delta E = h^2/(8m)(a^{-2} - b^{-2})$, where $a$ and $b$ are the length and the width of the box, respectively. Minor deviations from this dependence in Fig. 2c reflect the departure of the actual confinement potential from an idealised infinite rectangular well.

## Polariton spin-polarised far-field emission

Building on the characterisation of the polariton lasing threshold and energy splitting, we now turn to momentum-space characterisation to investigate the band structure of the elliptical micropillar chain. In the micro-photoluminescence setup, the far-field emission of the sample is investigated by imaging the back focal plane of the objective. This allows for direct access to momentum space, where the periodic nature of our structures enables the formation of band structures. Circular polarisation-resolved measurements are performed to reveal polariton-pseudospin-dependent effects.

The experimental band structure, measured below the lasing threshold using large-spot, off-resonant laser excitation that spans several unit cells, is shown in Fig. 3a at negative detuning between the quantum well exciton and the cavity mode (excitonic fraction of 3.5–5%). The six lowest-energy bands correspond to $s$-bands, formed from the fundamental $s$-like modes of the individual elliptical

micropillars, and are the primary focus of our study. The difference between the circular polarisation resolved band structures $I_{\sigma+} - I_{\sigma-}$ is shown in Fig. 3(b), highlighting how eigenstates at opposing wave-vectors ($+k_x$ and $-k_x$) have opposite circular polarisations at a given energy, analogous to the edge localisation presented in the band structure of our tight-binding Hamiltonian. These bands govern the topological and polarisation-dependent behaviour of the system and show excellent agreement with numerical simulations based on the Gross-Pitaevskii equation (GPE), as seen in Fig. 3c. We note a slight discrepancy between the experimental and theoretical band structures, which we believe arises from the simplification of treating the polarisation splitting as spatially uniform. Nevertheless, both theory and experiment consistently show the features of the topological spin Hall effect. Due to mirror symmetry-breaking of the non-ideal experimental geometry, the polarisation-resolved band structure is symmetrized around $k_x = 0$ (details in the methods section).

Numerical simulations of the band structure for a periodic lattice (Fig. 3d) reproduce the observed polarisation effects. Due to experimental reasons, the circular polarisation detection is only valid close to $k_x = 0$. Therefore, we limit the polarisation analysis of the far field to the first Brillouin zone of the system.

## Circular polarisation dependent directed propagation

Building on the insights gained from far-field band structure measurements, we now investigate the behaviour of polaritons in the micropillar chain under high excitation power. As in the momentum-space measurements, polarisation-resolved studies are employed to reveal polariton-pseudospin-dependent propagation effects. When the small-spot laser excitation power exceeds $P_{thr}$, polariton lasing occurs within the exciton-polariton micropillar chain. By again employing a negatively detuned chain with a high photonic fraction, we achieve polariton lasing in the third band of the chain. This lasing occurs in a state that has non-zero group velocity, enabling circular polarisation-dependent energy transport.

By analysing the difference between the $\sigma_+$ and $\sigma_-$ polarisation-resolved amplitudes (square root of measured intensity), the opposing propagation directions of the two polariton pseudospin states become

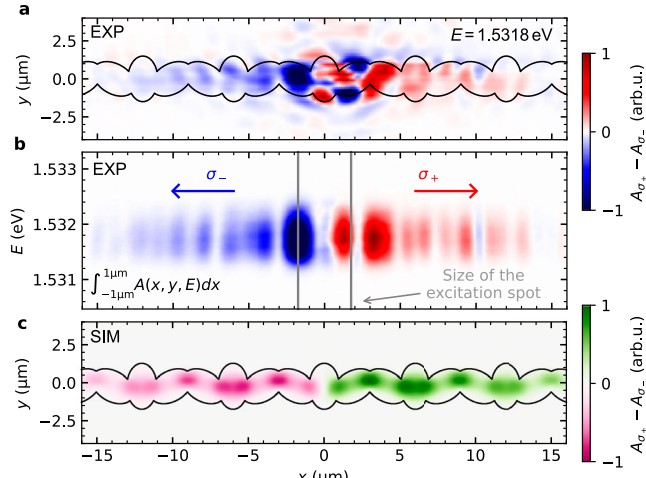

**Fig. 4 | Circular polarisation resolved real-space propagation of the exciton-polariton Bose–Einstein condensate. a** Constant energy cut through the lasing mode, showing the spatial distribution of the polariton condensate within the elliptical micropillar chain. The perimeter of the elliptical micropillars is outlined in black. The difference between the circular polarisation resolved spectra shows the polarisation-dependent propagation direction of the exciton polaritons within the chain. Light collected outside the chain is subject to non-polarisation-preserving scattering processes. **b** Energy-dependent polarisation, integrated around $y = 0$, showing a strong relation between circular polarisation and propagation direction. **c** Numerical simulation of the propagation dynamics, obtained by solving the driven-dissipative Gross–Pitaevskii equation.

evident. The real-space propagation measurement was performed at a detuning with a slightly shorter cavity length (slightly larger $k_z$ in Fig. 2a) and thus, a marginally higher excitonic fraction of the lower polariton, compared to the band structure measurements in Fig. 3. Besides enabling lasing in a higher energy state of the system, a high photonic fraction also leads to increased propagation lengths. In this configuration, at an excitonic fraction of 10.4%, we achieve a propagation length spanning several unit cells while lasing in the third band of the system visible in Fig. 3. A constant energy cut at $E_{topo}^{\sigma\pm} = 1.5318$ eV of the mode tomography along the chain shows the lasing mode (Fig. 4a). The $y$ integrated tomography in Fig. 4b, where integration is performed for light collected from within the chain, emphasises the polarisation-dependent propagation. Significant intensity in Fig. 4a is also observed outside of the micropillars, corresponding to light that is scattered in the sample with non-polarisation-conserving processes in the BCB. The experimental findings are consistent with numerical simulations based on the GPE depicted in Fig. 4(c).

## Discussion

In summary, we have experimentally realised the topological pseudospin Hall effect in a one-dimensional chain of elliptical micropillars. In this Hofstadter ladder scheme, we show polariton circular polarisation-dependent non-reciprocal transport. By introducing a synthetic gauge field through precise rotational alignment of the micropillar resonators, we have engineered a system where circular polarisation acts as an artificial dimension. This enables the implementation of topologically protected edge states that can propagate within a compact lattice geometry that is one-dimensional in real space.

Our approach demonstrates a significant reduction in spatial form factor compared to traditional two-dimensional exciton-polariton topological insulators that also feature chiral propagation, while preserving the defining features of topological protection. This compact design facilitates unprecedented control over the selective transport of polaritons based on their pseudospin (circular polarisation), showcasing potential for polariton circular polarisation-selective optically active

devices. These results pave the way for advanced polaritonic technologies, such as topological polariton transistors or logic gates that use circular polarisation as a logical bit, combining compactness with the inherent advantages of topologically robust exciton-polariton dynamics.

While our work strongly and precisely relies on synthetic gauge fields to achieve polarisation-selective propagation behaviour, growing efforts are establishing perovskite-based platforms for room-temperature exciton-polariton spin generation and the advancement of spin-optoelectronic technologies.[32,33] In contrast to these works, we utilise precise control over the photonic potential landscape, resulting in the observed effects in conjunction with polariton condensation under non-resonant excitation schemes.

This work opens exciting avenues for the exploration and realisation of artificial gauge fields in photonic systems by precisely engineering the photonic potential landscape. Such advancements enable the experimental implementation of non-zero gauge fields and artificial dimensions in exciton-polariton lattices, extending beyond the need for external magnetic fields. The inclusion of gain in our system allows for selective mode competition where only one mode emerges as the dominant lasing state. Our platform also provides a promising avenue for exploring non-Hermitian physics, particularly the non-Hermitian skin effect, by implementing differential decay rates for the two circular polarisation states[28,31]. Furthermore, while our current platform demonstrates one-dimensional topological edge states within a micropillar chain, the concept is generalisable to lattices that are two-dimensional in real space. Extending this approach to higher-dimensional lattices could facilitate the realisation of two-dimensional in-gap topological surface states in effective three-dimensional exciton-polariton systems with artificial dimensions. This is a critical step toward achieving high-power, coherent, topological surface-emitting lasers.

## Methods
### Experimental setup
For laser excitation of the microcavity, we employ a wavelength-tunable continuous-wave laser, while a white light source (tungsten–halogene) was used for reflection measurements. Excitation and detection were conducted through the same microscope objective, with a ×20 objective ($NA = 0.40$) for $k$-space measurements and a ×50 objective ($NA = 0.42$) for real-space measurements, operating in reflection geometry.

Polarisation detection was performed in the detection path using rotatable quarter-wave and half-wave plates placed in front of a linear polarizer. In the off-resonant laser excitation scheme, the photoluminescence from the sample was separated from the excitation laser using a long-pass filter. For wavelength-resolved measurements, a Czerny-Turner spectrometer (Andor Shamrock 750) was paired with a charge-coupled device (Andor iKon-M).

Mode tomography and hyperspectral imaging ($x$, $y$, $E$ or $k_x$, $k_y$, $E$) were performed by scanning the imaging lens across the entrance slit of the spectrometer. The sample was mounted in vacuum in a Janis liquid helium flow cryostat with an operating temperature of around 5 K for all photoluminescence measurements carried out in this work.

### Microcavity growth
The strong-coupling microcavity was grown using molecular beam epitaxy (MBE), ensuring precise control over the layer thicknesses and material compositions. The structure consists of a $\lambda/2$ cavity layer sandwiched between distributed Bragg reflectors (DBRs) composed of alternating AlAs and $Al_{0.15}Ga_{0.85}As$ layers. The bottom mirror contains 40 pairs, while the top mirror consists of 36 pairs, as shown in Fig. 5a. Each DBR layer has an optical thickness of $\lambda/4$. The $\lambda/2$ cavity layer as well as the first mirror layers incorporate four stacks of 13 nm GaAs quantum wells (QWs) embedded in 4 nm $Al_{0.88}Ga_{0.12}As$ barriers (Fig. 5b). This design ensures that the quantum wells are positioned at the maxima of the electric field intensity, optimising the overlap between the cavity photon and the quantum well exciton for achieving

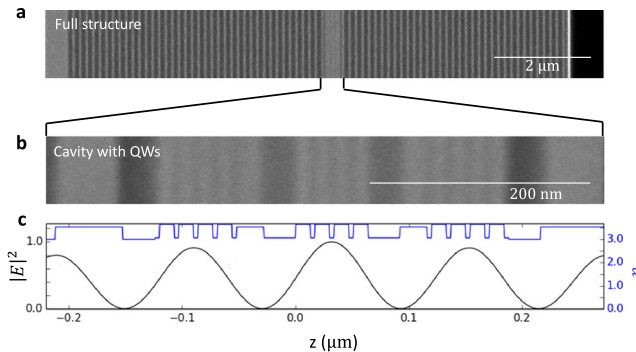

**Fig. 5 | Cross section of the microcavity structure. a** Scanning electron microscope image of the complete microcavity containing 40 bottom and 36 top AlAs/$Al_{0.15}Ga_{0.85}$As distributed Bragg reflectors. The alternating layers have an optical length of $\lambda/4$ and alternating refractive index. **b** Closeup view of the $\lambda/2$ cavity layer containing $4 \times 13$ nm GaAs quantum wells in 4 nm $Al_{0.88}Ga_{0.12}$As barrier. Two more quantum well stacks are positioned in the mirror pairs surrounding the cavity. **c**, Position-dependent electric field intensity of the cavity mode at zero incidence angle and material-dependent refractive index.

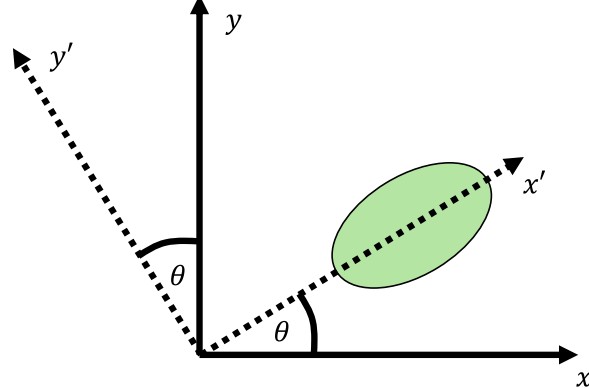

**Fig. 6 | Raw, unsymmetrized spectrum.** Circular polarisation resolved far-field emission of the elliptical micropillar chain (same measurement as Fig. 3b).

strong coupling. Fig. 5c illustrates the position-dependent electric field intensity of the cavity mode at normal incidence, highlighting the spatial overlap of the electric field with the quantum wells. The refractive index profile of the materials in the cavity and mirrors is also shown, emphasising the periodic structure critical for the DBR's performance.

## Microcavity patterning

The epitaxially grown microcavity wafer undergoes a series of processing steps to create the desired micropillar structures. A cleaved piece of the wafer, chosen for its appropriate detuning between the cavity mode and quantum well exciton, is spin-coated with an electron-sensitive photoresist. Using electron beam lithography, the photoresist is exposed, imprinting the targeted micropillar shape onto the resist. Following exposure, the exposed resist is removed, leaving behind the unexposed photoresist as a negative template. Barium fluoride and chromium are then deposited as a hard mask to provide robustness during the subsequent etching process. The remaining photoresist is removed chemically, leaving the bare microcavity with a protective hard mask on the regions where etching is not desired. Inductively coupled plasma etching is used to remove the unmasked regions of the microcavity. This technique achieves high aspect ratios of etch depth to feature size and smooth sidewalls. The etching continues until several layers of the bottom DBR are removed. After etching, the structure is spin-coated with benzocyclobutene as a protective polymer layer, followed by plasma ashing. Finally, the barium fluoride/chromium mask is removed using water.

For the measurements presented in this work, elliptical chains with a semi-major axis diameter of $3\,\mu m$ and a semi-minor axis diameter of $1.67\,\mu m$ are used. The unit cell consists of three elliptical micropillars, with their positions and orientations optimised through experimentation. Specifically, we study two types of configurations to tune the energy splitting: elliptical pairs with rotation angles of −60°, +60° and 0°, +60°. To ensure equivalent inter-ellipse coupling ($J$ in equation (1)), we adjust the spatial overlap of the ellipses such that the additional energy splitting due to hopping is identical for both configurations. This optimisation ensures that the real inter-ellipse hopping $J$ in equation (1) is equivalent for −60°, +60° and 0°, +60° degree coupled ellipses, critical for achieving the desired topological properties.

## Band structure symmetrisation

The experimental setup introduces asymmetries in the intensity of the measured far-field images at opposing angles. Such asymmetries

can be caused by an imperfect microscope objective, a slight angular misalignment of the sample $x$–$y$-plane normal vector with the optical axis of the detection path (sample gluing process), or etaloning on the CCD chip, which varies the measured intensity locally due to interference effects. The measurements presented in this work that are most affected by this effect are difference spectra in momentum space, such as the one presented in Fig. 3b. The unsymmetrized circular polarisation resolved difference spectrum of this dataset is presented in Fig. 6. While the qualitative circular polarisation dependence of the band structure is clearly visible, the measurement shows slight asymmetry with respect to $k_x = 0$. This imbalance can also be observed in a planar microcavity, where a physical asymmetry originating from within the sample is rigorously absent, confirming its instrumental origin. To emphasise the polarisation dependence in the main text, we symmetrise the spectrum shown in Fig. 6 by superimposing a mirrored spectrum (mirrored around $k_x = 0$).

## Basis transformation for complex hopping

Following the description of Mandal et al.[31], the complex phase in the hopping between the s-modes of an elliptical micropillar can be deduced by a basis transformation. Figure 7 considers an elliptical micropillar rotated by the angle $\theta$. The two s-modes are polarised along the semi-major and minor axis ($x'$, $y'$) of the ellipse with an energy splitting $\Delta_T$. In the diagonal ($x'$, $y'$) basis, the system dynamics are given by

$$i\hbar \frac{\partial}{\partial t} \begin{pmatrix} \psi'_x \\ \psi'_y \end{pmatrix} = \begin{pmatrix} \epsilon'_x & 0 \\ 0 & \epsilon'_y \end{pmatrix} \begin{pmatrix} \psi'_x \\ \psi'_y \end{pmatrix}. \tag{2}$$

The energy splitting is thus given by $\Delta_T = \epsilon'_x - \epsilon'_y$. Going to the basis $(x, y)$ can be achieved by applying a rotation matrix

$$\begin{pmatrix} \psi'_x \\ \psi'_y \end{pmatrix} = \begin{pmatrix} \cos\theta & \sin\theta \\ -\sin\theta & \cos\theta \end{pmatrix} \begin{pmatrix} \psi_x \\ \psi_y \end{pmatrix}. \tag{3}$$

Furthermore, the circular polarisation basis is given by

$$\begin{pmatrix} \psi_{\sigma_+} \\ \psi_{\sigma_-} \end{pmatrix} = \frac{1}{\sqrt{2}} \begin{pmatrix} 1 & i \\ 1 & -i \end{pmatrix} \begin{pmatrix} \psi_x \\ \psi_y \end{pmatrix} \tag{4}$$

and inversely

$$\begin{pmatrix} \psi_x \\ \psi_y \end{pmatrix} = \frac{1}{\sqrt{2}} \begin{pmatrix} 1 & 1 \\ -i & i \end{pmatrix} \begin{pmatrix} \psi_{\sigma_+} \\ \psi_{\sigma_-} \end{pmatrix}. \tag{5}$$

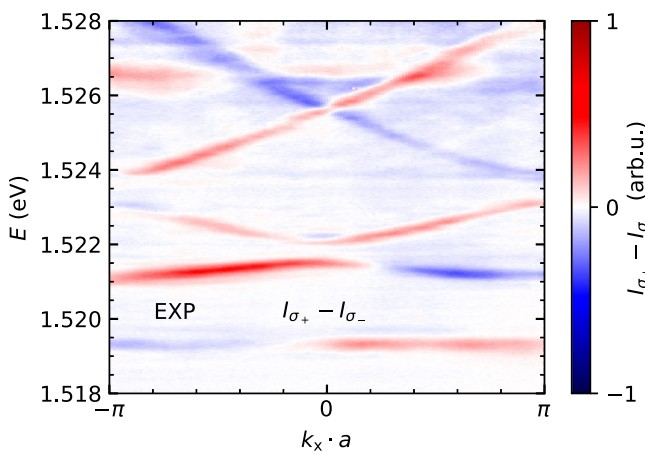

**Fig. 7 | Complex hopping basis transformation.** Elliptical micropillar rotated by the angle $\theta$. The two s-modes with lifted degeneracy are linearly polarised in the rotated basis $(x', y')$.

Thus, the circular polarisation basis is given by the transformation $T$

$$T\begin{pmatrix}\psi'_x\\\psi'_y\end{pmatrix}=\begin{pmatrix}\psi_{\sigma_+}\\\psi_{\sigma_-}\end{pmatrix}=\frac{1}{\sqrt{2}}\begin{pmatrix}1 & i\\1 & -i\end{pmatrix}\begin{pmatrix}\cos\theta & -\sin\theta\\\sin\theta & \cos\theta\end{pmatrix}\begin{pmatrix}\psi'_x\\\psi'_y\end{pmatrix} \quad (6)$$

$$=\frac{1}{\sqrt{2}}\begin{pmatrix}e^{i\theta} & ie^{i\theta}\\e^{-i\theta} & -ie^{-i\theta}\end{pmatrix}\begin{pmatrix}\psi'_x\\\psi'_y\end{pmatrix} \quad (7)$$

and inversely

$$T^{-1}=\frac{1}{\sqrt{2}}\begin{pmatrix}e^{-i\theta} & e^{i\theta}\\-ie^{-i\theta} & ie^{i\theta}\end{pmatrix}. \quad (8)$$

Using this, the Hamiltonian in the primed basis $H'=THT^{-1}$ reads

$$THT^{-1}=\frac{1}{2}\begin{pmatrix}\epsilon'_x+\epsilon'_y & e^{2i\theta}(\epsilon'_x-\epsilon'_y)\\e^{-2i\theta}(\epsilon'_x-\epsilon'_y) & \epsilon'_x+\epsilon'_y\end{pmatrix} \quad (9)$$

$$=\begin{pmatrix}\epsilon'_x+\epsilon'_y & e^{2i\theta}\Delta_T\\e^{-2i\theta}\Delta_T & \epsilon'_x+\epsilon'_y\end{pmatrix}. \quad (10)$$

Therefore, a rotation by an angle $\theta$ results in a complex phase factor with an angle of $2\theta$, requiring rotation angles of the ellipses by 60° to achieve an effective phase of $2\pi/3 = 120°$.

**Numerical Simulation**

To model the dynamics of exciton-polaritons, we use the Gross-Pitaevskii (GP) equation in the presence of a non-resonant pump:

$$i\hbar\frac{\partial\psi_{\sigma_\pm}}{\partial t}=\left[-(1-i\Lambda)\frac{\hbar^2\nabla^2}{2m_p}+V(x,y)-i\Gamma\right]\psi_{\sigma_\pm}+V_T\psi_{\sigma_\mp}+(g+iP_0)P(x,y)\psi_{\sigma_\pm}$$
$$+(\alpha_1-i\alpha_{NL})|\psi_{\sigma_\pm}|^2+\alpha_2|\psi_{\sigma_\mp}|^2\psi_{\sigma_\pm} \quad (11)$$

Here $\psi_{\sigma_\pm}$ represents the polariton wavefunctions corresponding to $\sigma_\pm$ spins. The term involving $\nabla^2\equiv(\partial^2/\partial x^2+\partial^2/\partial y^2)$ accounts for the kinetic energy of the polaritons with mass $m_p$. $\Lambda$ represents the energy relaxation term. $V(x,y)$ is the potential profile of the elliptical micropillar chain, and $\Gamma$ represents the linear decay due to the finite lifetime of the polaritons. The term $V_T(x,y)$ describes the polarisation splitting inside the micropillars, which has the same spatial profile as $V(x,y)$ but is multiplied by $\exp(\pm2i\phi_n)$, where $\phi_n$ is the orientation angle of the

n-th pillar. The terms $g$ and $iP_0$ represent the potential and gain induced by the excitonic reservoir, created by the non-resonant pump $P(x,y)$. $\alpha_1(\alpha_2)$ characterises the polariton-polariton interactions with the same (opposite) spin, while $\alpha_{NL}$ is the nonlinear loss term, an inevitable consequence of the non-resonant pump. In this model, we assume the dynamics of the excitonic reservoir is faster compared to the polariton dynamics, an assumption typically consistent with experiment.

To study the band structure of the system, we first remove the pump, decay, and nonlinear terms from the above equation. By numerically diagonalizing the linear Hermitian system consisting of 15 unit cells and Fourier transforming the eigenstates, the band structure shown in Figure 3c is obtained. Alternatively, applying Bloch's theorem to the unit cell yields the band structure in 3d, where each state is colour-coded according to its spin contribution, showing that different spins have opposite group velocities. Both band structures agree with the experimentally measured ones in Figs. 3a, b. We used a polariton mass $m_p = 2.75\cdot10^{-5}m_e$, where $m_e$ is the free electron mass. The potential depth was set to 100 meV, and the effective polarisation splitting was 4 meV. To account for possible variations in the size of the polarisation splitting between pillars, the splitting for pillars with $\phi_n = \pi$ was taken as one-third of that in the other two pillars. This is an approximation of the experimental reality, so minor differences between experiment and theory remain expected. Finally, to obtain the polariton dynamics under the non-resonant pump, we reintroduce the pump, decay, and nonlinear terms in the GP equation. We note that while the dynamics of the polaritons are primarily governed by the linear terms, we have included the nonlinear inter-action terms for completeness. The non-resonant pump is modelled as a Gaussian:

$$P(x,y)=\exp\left(-\frac{(x-x_0)^2+(y-y_0)^2}{2\Delta^2}\right), \quad (12)$$

where $(x_0, y_0) = (0, 0)$ is the pump position, and $\Delta = 2.8\,\mu m$ is its width. Using an initial random condition, we simulate the system with the GP equation and obtain the steady state, which is shown in Fig. 3c. This result aligns well with the experimental data in Figs. 3a, b. For simplicity, the strength of polarisation splitting was kept constant across all pillars for this calculation. Some small differences in the real space pattern are observed in the vicinity of the laser spot; however, these are expected. The laser should undergo some level of diffraction caused by the three-dimensional photonic structure, which are not accounted for in the two-dimensional Gross-Pitaevskii simulations that assume that the light has a Gaussian distribution when reaching the exciton layer. The decay rate is $\Gamma = 6.6\,\mu eV$, corresponding to a polariton lifetime of 50 ps. Other parameters are: $\Lambda = 5\cdot10^{-3}$, $g = 1$ meV, $P_0 = 6\Gamma$, $\alpha_1 = 1\,\mu eV\cdot\mu m^2$, $\alpha_2 = -0.1\,\alpha_1$, and $\alpha_{NL} = 0.3\,\alpha_1$. All other parameters remain the same as in Fig. 3c, d.

**Robustness analysis**

To test the robustness of the topological modes, we introduce a coherent pump in our numerical simulations, governed by the following GP equation:

$$i\hbar\frac{\partial\Psi_{\sigma_\pm}}{\partial t}=\left[-\frac{\hbar^2\nabla^2}{2m_p}+V(x,y)-i\Gamma\right]\Psi_{\sigma_\pm}+V_T(x,y)\Psi_{\sigma_\mp}$$
$$+F_{\sigma_\pm}(x,y)\exp[ik_{p_{\sigma_\pm}}-\omega t]. \quad (13)$$

Here $F_{\sigma_\pm}$ represent the coherent pump having energy $\hbar\omega$ and wave vector $k_{\sigma_\pm}$. Since the excitation is coherent, the energy relaxation term is neglected. We also assume a low pump power regime, allowing us to ignore nonlinear interactions. Under these conditions, we

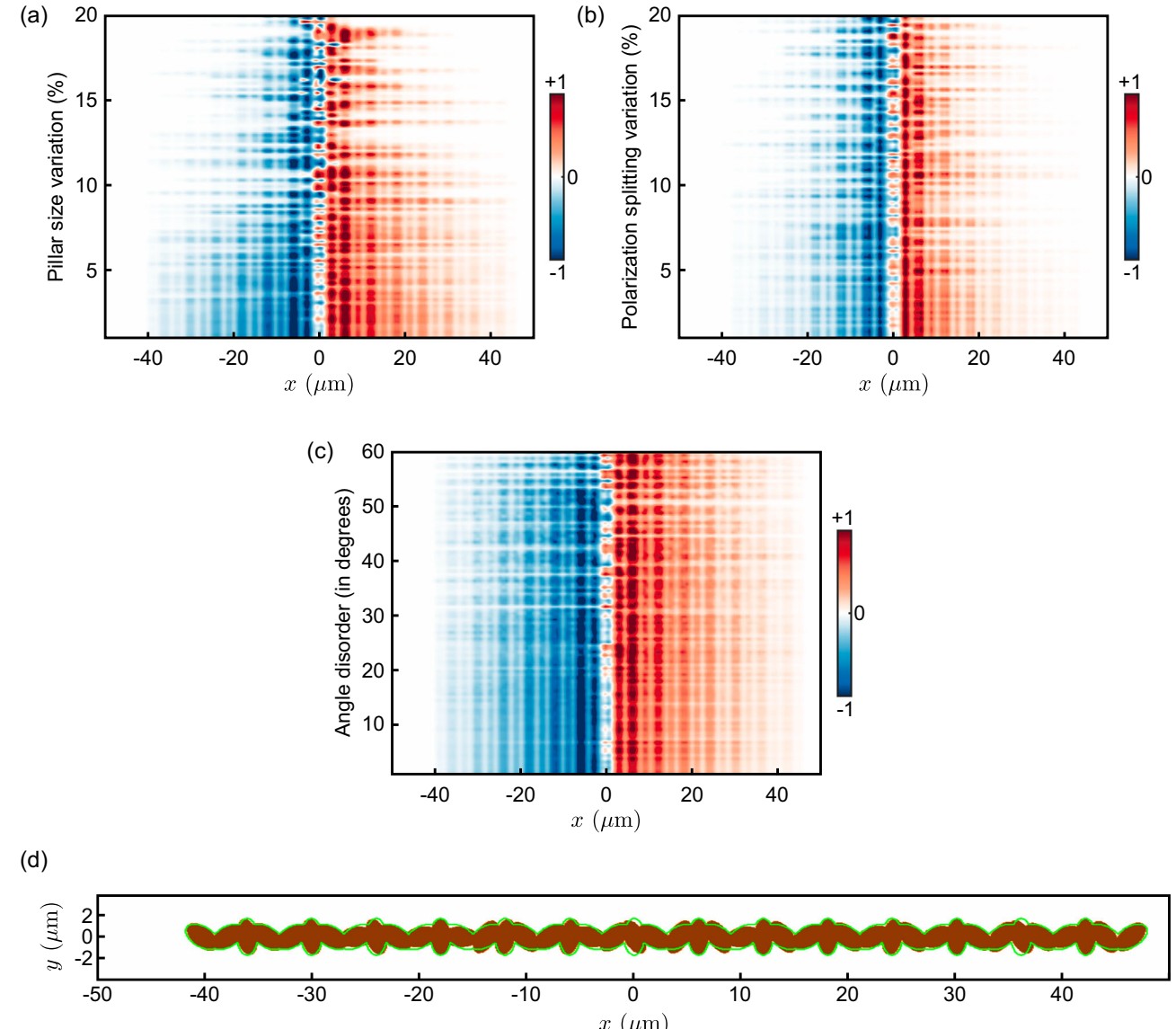

**Fig. 8 | Robustness of topological modes. a–c** Polariton spin propagation under different types of disorder: (**a**) random variations in pillar sizes, (**b**) random variations in polarisation splitting, and (**c**) random disorder in pillar orientation angles. **d** Visualisation of the strongest disorder in pillar orientation (brown) from the ideal orientation (green). Despite increasing disorder, the spin-polarised propagation remains largely robust. In all the above calculations, the energy of the coherent pump was fixed at 1.522 eV.

position the pump at the centre of the chain and compute the steady-state solution. From this steady state, we extract the degree of circular polarisation and integrate it over the width of the chain and define the following quantity to characterise the robustness:

$$I(x) = \int \left( |\Psi_{\sigma_+}(x,y)|^2 - |\Psi_{\sigma_-}(x,y)|^2 \right) dy. \qquad (14)$$

Next, we deliberately introduce random variations in the size, polarisation splitting, and orientation angles of the micropillars, and compute $I(x)$ for each case. The random variations are introduced by adding uniformly distributed random values to the disorder-free structure. As shown in Fig. 8, even in the presence of significant disorder in the pillar sizes (panel a) and polarisation splitting (panel b), the spin propagation remains intact. However, as expected, the propagation distance decreases with increasing disorder strength as localisation effects kicks in.

In contrast, the spin propagation remains largely unaffected by disorder in the orientation angles of the pillars. This highlights the role of the artificial gauge flux in our system: variations in orientation angles alter the local flux associated with each plaquette. However, since our scheme is tied to the Hofstadter butterfly spectrum, which supports a non-zero Chern number in the lowest bandgap for all flux values $0 < \phi < \pi$, the spin-polarised edge states persist even under substantial angular disorder.

Therefore, the topological origin of these modes ensures robust spin propagation in the presence of various types of disorder. We emphasise that the disorder levels considered here exceed those currently achievable in state-of-the-art polariton lattices.

## Data availability

The data generated in this study have been deposited in the wuedata database [wuedata.uni-wuerzburg.de]. The data is listed under this publication link: https://doi.org/10.58160/xfkn48dhnj1gbt2g.

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

## Acknowledgements

The Würzburg group acknowledges financial support by the German Research Foundation (DFG) under Germany's Excellence Strategy-EXC2147 "ct.qmat" (project id 390858490) and within the project KL3124/3.1. T.C.H.L. was supported by the Singapore Ministry of Education (MOE) grant (MOE-MOET32023-0003) "Quantum Geometric Advantage".

## Author contributions

S.W., J.B, J.D, C.G.M. and S.B. built the experimental set-up, performed the experiments and analysed the data. S.D. and P.G. grew the samples by molecular beam epitaxy. M.E., S.W. and J.B. realised the layout, etching and nanofabrication of the samples. S.M., R.B., T.C.H.L. and R.T. realised the theoretical calculations and numerical simulations. All authors participated in the scientific discussions about all aspects of the work. S.W. and S.K. wrote the original draft of the paper. All authors reviewed and edited the paper. T.C.H.L. and S.K. conceived the idea. S.H. and S.K. supervised the work.

## Funding

## Competing interests

The authors declare no competing interests.
