## [Transparent Peer Review file · Nature Communications]

Artificial Gauge Fields and Dimensions in a Polariton Hofstadter Ladder

Corresponding Author: Professor Sebastian Klemmt

Version 0:

Reviewer comments:

Reviewer #1

(Remarks to the Author)

The authors have satisfactorily addressed my comments. I am happy to recommend publication of this work in Nature Communications.

Reviewer #2

(Remarks to the Author)

In the revised version, the authors have addressed all of my comments, and I am satisfied with most of them. However, regarding my main concern about the novelty, the response letter emphasizes that forward and backward propagating states carry opposite circular polarizations due to a non-reciprocal artificial gauge field, which they argue distinguishes their work from existing literature. I would like to point out that similar observable signatures have been reported in recent polaritonic systems.

In particular, Liang et al. (Nat. Photon. 18, 357–362 (2024). <https://doi.org/10.1038/s41566-023-01375-x>) demonstrated the polariton spin Hall effect in perovskite–liquid-crystal microcavities operating in the Rashba–Dresselhaus regime. Synthetic spin–orbit coupling produced spin-split bands with opposite circular polarizations in different momentum-space valleys, leading to permanent spin separation with high purity (~0.9) over tens of microns. I note that their Fig. 2f (showing spin-split circularly polarized bands) appears qualitatively very similar to Fig. 3b of the present manuscript in terms of band structure. Likewise, Wang et al. (Advanced Materials <https://doi.org/10.1002/adma.202412952>) realized electrically generated spin-polarized polaritons in perovskite metasurfaces. By breaking both in-plane and out-of-plane inversion symmetries, they achieved a Rashba-like band splitting where forward and backward propagating polaritons carried opposite spins, with electrical bias enabling selective directionality. Although the physical mechanisms in these works are based on reciprocal Rashba/Dresselhaus-type spin–orbit couplings, the resulting phenomenology—namely, opposite circular polarizations associated with opposite propagation directions—appears closely related to what the present manuscript reports.

While I find the present work interesting and original, before I can recommend its publication in Nature Communications, the authors should provide a clear and critical discussion of how their proposed non-reciprocal gauge field leads to genuinely distinct band-structure physics, and what specific advantages it offers over these existing implementations (e.g., robustness, scalability, or new transport behaviors).

Version 1:

Reviewer comments:

Reviewer #2

(Remarks to the Author)

The authors addressed my concerns in the revised manuscript and the response letter. I am pleased with their responses, and now recommend the publication of the work in Nature Communications.

Subject: Response to Referees regarding “Artificial Gauge Fields and Dimensions in a Polariton Hofstadter Ladder” for publication in Nature Communications

Point-by-point Reply

Referee #1

We thank the referee for accepting the changes to our revised version.

Referee #2

1. Referee #2 refers to a paper by [Liang et al., *Nat. Photon.* 18, 357–362 (2024)], where the authors demonstrated the polariton spin Hall effect in perovskite–liquid-crystal microcavities operating in the Rashba–Dresselhaus regime.
2. The referee refers to a paper by [Wang et al., *Adv. Mater.* 37, 2412952 (2025)], where the authors realized electrically generated spin-polarized polaritons in perovskite metasurfaces.

Regarding [Liang et al., *Nat. Photon.* 18, 357–362 (2024)]:

Although the band structure is spin-polarized due to the Rashba-Dresselhaus effect, spin propagation is not universal – the experiments strongly require resonant pumping and low disorder. As shown in Fig. 2f, for a given energy there typically exist both right- and left-propagating states for each spin. Without carefully tuned excitation, or in the presence of disorder, these states can mix, destroying the spin-Hall effect signature. This likely explains why a resonant pump was used in that work. Under non-resonant pumping, where momentum selection is generally difficult, the scheme is not expected to perform as effectively.

In contrast, in our work the left- and right-propagating states carry opposite spins across most of the energy window. Consequently, even if disorder couples counter-propagating states, the spin must necessarily flip. This makes spin propagation in our system inherently more robust and helps explain why we observe unequivocal spin transport even under non-resonant pumping conditions.

Moreover, our design and experiment introduce a locally controllable artificial gauge field: the orientation of each elliptical pillar defines the preferred propagation direction for a given spin. In principle, this allows different preferred directions to be engineered at different locations within the same sample – a feature that could be highly relevant for applications in future polaritonic chips and therefore strongly highlighting the novelty and importance of our approach.

Regarding [Wang et al., *Adv. Mater.* 37, 2412952 (2025)]:

This interesting work represents a rather different setup, where the characteristic advantages of microcavity polaritons – such as strong interactions and high Q-factors – are not fully present. To the best of our knowledge, condensation has not yet been demonstrated in this platform, including in the specific work under discussion. Consistent with this, their abstract concludes with the statement:

“Here, the development of compact and tunable spintronic devices is advanced and an important step toward the realization of electrically pumped inversionless spin-lasers is represented.”

In this sense, apart from the aspect of electrical injection, our work already realizes what they describe as a very “important step.”

It also appears that their study does not directly address propagation. The angle-resolved measurements show that states can be excited in different directions, but the extent of propagation is not fully clear. Their emphasis, therefore, is on generating circularly polarized states via electrical injection, which is a valuable but different objective from our focus on controlling directional spin propagation along a channel or chain of pillars.

Finally, our work is employing well-established III/V semiconductors as opposed to the use of perovskite materials. We believe it is still an open question which material platform is best suited for polaritonic devices, and as of right now developing different materials is very worthwhile. E.g. it is not obvious that liquid crystals could induce a Rashba-Dresselhaus Hamiltonian in a GaAs system.

In summary, we have no doubt that both works brought up by the respected Referee #2 are addressing very relevant aspects of polarization control on complex optoelectronic systems. We have added reference to these works in the main text of our paper to allow the reader a comprehensive overview. However, for the reasons detailed above, we are confident that our work represents an entirely novel and important step towards functional logic devices based on polariton pseudo-spin control and propagation.